# Microbiological and Physicochemical Approach in the Feeding of Superworm (*Zophobas morio*) with Petroleum-Derived Polymer Diets

**DOI:** 10.3390/microorganisms12112118

**Published:** 2024-10-23

**Authors:** Brandon R. Burgos, Fabiola Morales, Rodrigo Morales-Vera, Cristian Valdés, Jorge Y. Faundez, Eduardo Pereira de Souza, Flavio Henrique-Silva, Ariel D. Arencibia

**Affiliations:** 1School of Biotechnology Engineering, Center for Biotechnology of Natural Resources (CENBio), Catholic University of Maule, Avda. San Miguel 3605, Talca 3460000, Chile; brandon.burgos@alu.ucm.cl (B.R.B.); fabiola.morales@alu.ucm.cl (F.M.); 2Center for Biotechnology of Natural Resources (CENBio), Catholic University of Maule, Avda. San Miguel 3605, Talca 3460000, Chile; cvaldesv@ucm.cl; 3Maule Advanced Studies Research Center (CIEAM), Catholic University of Maule, Avda. San Miguel 3605, Talca 3460000, Chile; 4Doctorate in Translational Biotechnology (DBT), Catholic University of Maule, Avda. San Miguel 3605, Talca 3460000, Chile; jorge.faundez@alumnos.ucm.cl; 5Department of Genetics and Evolution, Federal University of São Carlos (UFSCar), São Carlos 13565-905, SP, Brazil; edupsouza96@gmail.com (E.P.d.S.); dfhs@ufscar.br (F.H.-S.)

**Keywords:** *Zophobas morio*, synthetic polymers, intestinal bacteria, SEM, FTIR

## Abstract

Plastics are very versatile materials that have contributed to the development of society since the 19th century; however, their mismanagement has led to an accumulation of plastic waste in almost every ecosystem, affecting the fauna of the planet. However, recently, some studies have shown that some insects might be able to adapt, consuming a wide range of hydrocarbon base polymers. In this work, the adaptive capacity of *Zophobas morio* larvae when feeding on different synthetic polymers derived from petroleum was studied. Four different thirty-day larval feeding treatments were carried out with synthetic polymers, including expanded polystyrene (PS), low-density polyethylene (LDPE), polyisoprene (PI), and butyl rubber (BR); in addition, a positive control of organic diet was included. Intestinal bacteria were isolated from the treatments and identified by Sanger sequencing. To analyze the chemical composition and physical form of the frass produced, Fourier transform infrared spectroscopy (FITR) was performed, and images of the feces’ surfaces were taken with scanning electron microscopy (SEM), respectively. *Zophobas morio* larvae were able to consume 54% of PS in 30 days, equivalent to 3.2 mg/d/larva. Nine culturable bacterial strains associated with the decomposition of synthetic polymers were identified in the intestine of the larvae. As for the physicochemical analysis of the feces, FTIR spectra showed the scission of bands corresponding to functional groups of the synthetic polymers in the comparison of the plastic diet treatments versus the feces of antibiotic-treated and plastic-fed larvae, while the comparison of spectra of the plastic and control treatments also identified differences in the absorption peaks. SEM imaging demonstrated that superworm feces differed in dependence on the substrate consumed. The findings demonstrated that *Zophobas morio* larvae possess a gut biological complex that allows them to feed and survive by consuming various petroleum-derived polymers.

## 1. Introduction

Plastic is one of the most widely used materials in the world today. It is used as a construction, automotive, and packaging material, among other applications, due to the synthetic polymers, which are durable, malleable, economical, and versatile [1], making plastic a fundamental part of our lives in the last century. The indiscriminate use of plastic generates a series of problems to ecosystems, especially in countries that produce major plastic waste, such as Australia, Western Europe, and the UK [2]. Since the Second World War, the production of plastic materials has increased, generating an accumulation of 8300 megatons (Mt), of which only 2500 mt was in use, 500 mt was recycled, and the rest was disposed in both landfills and the natural environment, affecting the different ecosystems [3]. In 2019, 460 Mt of plastics was generated [4]; the following year, due to the SARS-CoV-19, approximately 200 tons per day of plastic waste was produced in the city of Wuhan alone [5]. The effect of these polymers on the environment has produced an increase in research during the last four years, which has led to the design of various remediation technologies to avoid and decrease the impact generated by these wastes [6].

There are a multitude of methods designed for the elimination or removal of plastic materials in different environments, among which chemical, physical, and biological methods are recognized. The latter are the technologies most investigated by biotechnologists, since they include the use of living organisms to remediate contaminating compounds, in addition to being environmentally friendly and more economical in their application [7].

The biodegradation of plastic fragments is considered one of the imminent solutions to mitigate the environmental threat [8]. The difficult biodegradability of petroleum-based synthetic polymers hinders the use of living organisms in their management; however, during the last few years, potential microorganisms have been discovered for their decomposition, which use the plastic molecules as a carbon source [9], where microbes use their enzymatic complexes, allowing them to degrade the polymers in several stages and, thus, obtain energy from them [10]. This method produces modifications in the physicochemical properties of the plastic, converting it into by-products that are less polluting for the environment.

Currently, research on the biodegradation of plastics has focused on animal species, such as insects, specifically on holometabolous larvae, such as Coleoptera and Lepidoptera, which have been shown to be able to consume plastics and mineralize them thanks to their intestinal microbiota [11]. The intestinal microbiota is a fundamental “organ” in the life of insects, since this allows them to adapt and interact with the environment of the place where they live, because certain intestinal bacteria have ecological functions that directly influence the behavior and physiology of insects [12]. The darkling beetle *Zophobas morio* of the family Tenebrionidae belongs to the order Coleoptera and is an insect of high industrial potential due to its varied applications, since the larval stage of this organism has several characteristics that make it a potential focus of study [13]. The larvae of *Zophobas morio* have a high content of nutrients, such as proteins and fats, which has allowed them to be used to formulate food for livestock; however, these insects have been shown to be able to feed on some synthetic polymers and survive for long periods of time, with which it has been recognized that in their intestines, there are microorganisms with enzymes that have the ability to degrade synthetic polymers, allowing the superworms to obtain nutrients and energy [14]. Superworms are able to chew and feed on polystyrene foam. These polymers are biodegraded in their intestines, allowing them to feed and continue their lifecycle until they reach their adult stage of beetle, which does not consume plastic [15]. However, their ability to degrade other synthetic materials present in nature has not yet been investigated. Therefore, in this work, for the first time, reused plastics that are commonly used by the population, such as PI, LDPE, and BR, are used as a diet for superworms. *Zophobas morio* larvae have demonstrated that they can consume many types of substrates, so it is expected that they are able to feed and survive by basing their diet on these synthetic polymers. In addition, the objective of this work is to evaluate the metabolic capacity and recognize the intestinal bacteria of *Zophobas morio* larvae when fed with different synthetic polymers. The results obtained will have important value for understanding the biological complex of superworms that allows them to metabolize the plastic material, allowing the development of new research associated with bioremediation.

## 2. Materials and Methods

### 2.1. Experimental Design and Feeding Treatments

*Zophobas morio* larvae were provided by the company Grillos y más SpA (Pitrufquén, Chile) and experiments were conducted in the laboratories and greenhouse of the Center of Biotechnology in Natural Resources (CENBio), University Catholic of Maule, Chile. The feeding treatments of *Zophobas morio* larvae based on plastics derived from petroleum were the following: 5 g of PS—expanded polystyrene, or better known as Styrofoam, 5 g of LDPE—low-density polyethylene from plastic bags, 10 g of PI—polyisoprene, 10 g of BR—butyl rubber, and 10 g of a control—organic matter (mix of potato, carrot, and apple). Each treatment had 3 replicates using aluminum containers (length × width × height = 20 × 13 × 4 cm), with 30 larvae of 10-day-old per replicate. The treatments were incubated for 30 days in a microclimate chamber (RGX-400EF, FAITHFUL, Huanghua, China), which was regulated at 25 °C and 40% relative humidity. The variables determined were the following: number of live larvae after 30 days, survival rate (%), substrate consumption (%), consumption rate (mg/d/larvae), and weight gain (mg/survival larvae). The climatic chamber was in a double-door greenhouse to manage escape risks. The experiment was monitored every three days to hydrate larvae and remove dead maggots.

### 2.2. Statistical Analysis

The data measured in the feeding treatments were processed and analyzed using IBM SPSS Statistics version 25, with a predetermined significance level of *p* = 0.05 for all statistical evaluations. Normality tests were performed to determine if the data followed a normal distribution and homogeneity. Subsequently, ANOVA followed by a Tukey’s test was used to determine statistical significance among treatments.

### 2.3. Isolation of Bacteria from Gut of Zophobas morio Larvae

After 30 days of feeding with the different diets, a random sample of 1 larva was selected for each treatment. The larvae of each group were washed separately 3 times with sterile distilled water to control major contaminants. They were subsequently dissected under a binocular magnifying glass, making a cut on the lateral side from the head to the end of the abdomen to extract the intestine. Intestinal samples were stored in 15 mL centrifuge tubes. The larval intestine samples were mixed with 10 mL of 0.9% saline. To homogenize the samples, sterile beads were added during the vortex shaking for 10 min. Serial dilutions were performed with reference to the 0.5 McFarland standard. After this, 300 μL aliquots of the dilutions were taken to dispense into Petri dishes with LB Agar. The plates were incubated for 24 h at 30 °C. After 24 h, microbial colonies were isolated according to diverse morphologies and higher growth. For each isolate, three successive subcultures were performed.

### 2.4. Identification of Culturable Intestinal Bacteria of Zophobas morio

All DNA extractions of the bacteria isolated from the larvae of each feeding treatment were performed with a total DNA extraction kit (NucleoSpin Plasmid, Mountain View, CA, USA) following the protocol suggested by the manufacturer. PCR was performed to amplify the *16S* gene with universal primers (27F/1492R). PCR products were purified with an EXO+SAP enzymatic clean-up kit (Thermo Fisher Scientific, Waltham, MA, USA). Amplicons were quantified by agarose gel electrophoresis with a molecular weight marker. A sequencing reaction for the study of the V3–V4 regions of the 16S rRNA gene was performed using a BigDye kit (Omega Bio-Tek, Norcross, GA, USA), with the primers 314F: 5′-CCTACGGGGGNGGCWGCAG-3′, and 805R: 5′-GACTACHVGGGTATCTAATCC-3′. A final purification was performed with the BigDye X-terminator kit (Omega Bio-Tek, Norcross, GA, USA). Samples were sequenced with the 3130XL platform (Applied Biosystems, Waltham, MA, USA) using the Sanger method at the Central Analítica of the Instituto de Química of Universidade de São Paulo, Brazil. The sequences obtained were cleaned with Bioedit 7.7 (Nucleics Co., Sydney, Australia) and aligned with Mafft alignment software, version 7. The sequences were deposited in the BLAST (Basic Local Alignment Search Tool) database of the National Center for Biotechnology Information (NCBI) to identify the bacterial species according to the established parameters.

### 2.5. Microbial Role in the Ingestion of Plastics by Zophobas morio

In order to determine the microbial role in the ingestion of plastics by *Zophobas morio*, the susceptibility to antibiotics of the culturable strains isolated from the intestines of *Zophobas morio* was evaluated by antibiogram analysis. Streptomycin (STR), kanamycin (KAN), ampicillin (AMP), and neomycin (NEO) antibiotics were utilized at 100 ug/mL. The diameter of the halo of inhibition (mm) was used as a selection criterion for the best treatment. Subsequently, the *Z. morio* larvae were fed with a diet supplemented with antibiotics to control the major intestinal bacteria. In this pretreatment stage, a total of sixty larvae were fed with a mixture of organic substrate with 500 mg of ampicillin. After 10 days of organic feeding with the antibiotic supplement, the larvae were divided into 2 separate groups to be fed during an additional 15 days with the synthetic polymers that previously showed the best ingestion results (PS and LDPE). Each treatment had two replicates, and the samples were maintained under controlled conditions, as explained previously in Section 2.1.

### 2.6. Physicochemical Analysis of Frass

In order to identify the changes in the chemical structure and morphology of the polymers after the larvae digestion, 10 mg of frass was collected from the plastic-based diet treatments with polystyrene foam (PS) and polyethylene bags (LDPE), including *Zophobas morio* feces treated with and without antibiotics and a polymer control with no treatment. Samples were analyzed using FTIR with attenuated total reflectance (ATR; PerkinElmer, Waltham, MA, USA), available at the Bioprocess Laboratory, CENBio, of the Universidad Católica del Maule. In addition to verifying differences in the frass surfaces, samples of the feces of the PS treatment and organic control were studied using the scanning electron microscope (SEM; Zeiss EVO MA10, Oberkochen, Germany) of the electron microscopy unit (UME) at the Universidad Austral de Chile.

## 3. Results

### 3.1. Plastic Feeding/Consumption

In the first two days of the experiment, traces of bites and ingestion were observed on plastic materials used as food for the *Zophobas morio* larvae (Figure 1). From the third day onwards, the differences were more noticeable, with large traces of feeding (holes and galleries) evident in the PS treatment (Figure 1B). This plastic diet was the most consumed by the larvae among the petroleum derivatives studied. On the other hand, it was observed that the larvae also ingested material from LDPE bags (Figure 1C), while in the treatments with PI and BR, the visual traces of feeding were minimal (Figure 1D,E). According to what was expected in the organic diet as a control treatment (mix of potato, carrot, and apple), the *Zophobas morio* larvae consumed the entire substrate during the first 20 days of the study (Figure 1A).

The performance of *Z. morio* larvae was quantified after 30 days of feeding with the different diets (Table 1). Substrate consumption (%) and the consumption rate were significantly higher when the larvae were fed with PS, compared to the other synthetic diets studied (*p* < 0.05). In this case, the larvae consumed 52.4% of the substrate, with an increase in weight of 3.2 mg/d/larvae, values significantly higher than those obtained in the diet with LDPE, which were 4.2% and 0.25 mg/d/larvae, respectively. Low values of consumption parameters were observed on the diets based on LDPE and PI. However, the BR diet was the less attractive diet in terms of consumption and preference for *Zophobas morio* larvae. As expected, the control treatment of larvae fed the organic diet showed the best results, with significant differences for the variables substrate consumption (100%), consumption rate (11.8 mg/d/larvae), and survival rate (93.2%). It was noticeable that in both the control and PS treatments, some larvae were found in the pupal stage; consequently, they were counted as live larvae in the final monitoring. The larvae of the plastic diet treatments showed a weight loss when fed with synthetic polymers, obtaining a negative difference between the average initial weight of the larvae and the average final weight of the surviving superworms of each treatment.

### 3.2. Identification of Culturable Intestinal Bacteria of Zophobas morio

A total of fourteen culturable bacterial strains were analyzed by sequencing the 16S DNA region using the universal primers 27F/1492R (Table 2). The BLAST results showed that five strains were identified from the control treatment (organic diet), two strains from the PS synthetic diet, three strains from the LDPE diet, and four bacterial strains from the PI diet. In all cases, the percentage identity of the sequences ranged from 96% to 100%. It was noted that the low feeding efficiency of *Zophobas morio* with the BR diet caused difficulties with the quality of the intestinal dissections, so this treatment was discarded from the analysis.

The results demonstrated that the isolated bacteria changed among the synthetic diet treatments with respect to the organic control. Five culturable bacterial strains were identified in the larval gut of the control, and for the synthetic-based diets, four, three, and two strains were identified for the PI, LDPE, and PS treatments, respectively (Table 2). The culturable bacteria identified in the PS diets were *Enterobacter* sp. and *Hafnia* sp., which differed from the bacteria recognized in the LDPE treatment. The treatment with the PI substrate showed different bacterial strains than the other synthetic substrate diets. This finding could suggest that microbial abundance and diversity in the intestine of larvae might depend on the type of diet (organic or synthetic) they ingest. These bacterial groups might be related to the degradation of synthetic materials in the larval gut due to their substrate-dependent specificity to the substrate consumed by superworms.

### 3.3. Determining the Microbial Role in the Ingestion of Plastics by Zophobas morio

The antibiotics used formed halos of inhibition in the antibiograms of bacteria isolated from the intestines of larvae fed with synthetic polymers; however, ampicillin proved to be more effective in all cases, obtaining an average 24.4 mm-diameter halo of inhibition (Figure 2). Therefore, it was selected to further treat the larvae of *Zophobas morio*.

Frass from antibiotic- and non-antibiotic-treated larvae were used to elucidate differences in the potential polymer degradation by comparison of FTIR spectra. There were clear dissimilarities in the peaks obtained (Figure 3A), recognizing an attenuation of the functional groups of the plastic, confirming that the intestinal bacteria of the larvae produced changes in the digestive process of the Styrofoam. Differences in degradation were observed between the fecal samples of larvae fed with LDPE and the fecal samples of superworms treated with antibiotics (Figure 3B), since there was a cleavage of some representative peaks of LDPE and the addition of new spectra corresponding to the oxidation of the material.

The samples analyzed with the scanning electron microscope showed differences in the morphology of the feces. It can be seen that the samples were quite similar, but in some areas, they presented different structures, since in the feces from the PS treatment, residues with hexagon shapes were observed, in addition to bacillus-type bacteria adhered to the surface (Figure 4A,B), while in the organic matter control treatment, a more homogeneous shape of the feces was observed, with spherical structures surrounding the entire surface (Figure 4C,D).

## 4. Discussion

Plastic materials have posed a significant challenge to the community, so finding strategies to mitigate the negative effects of their recalcitrant degradation is essential. In this study, relevant information on the adaptation of superworms was obtained by subjecting them to a diet based on commonly used synthetic polymers. The larvae of *Zophobas morio* were able to survive feeding on synthetic polymers thanks to their digestive tract and the microbes present in it, which allow the plastics to be degraded and mineralized. These insects had considerably good survival rates, with BR being the plastic that had the lowest performance, with 10% survival, which is why it was discarded for the following analyses. The PS obtained 53.5% survival after 30 days of the feeding trial, a result considered lower than in the case of other previous reports [15]. The average PS consumption rate was calculated to be 3.2 mg/d/larvae, a result considerably higher than the 0.43 mg/d/larva obtained previously [16]. In the LDPE-based diet treatment, a survival rate of 67.4% and a consumption rate of 0.2 mg/d/larvae were obtained. Comparable results were previously obtained, where under similar conditions they managed to obtain a survival rate of 94% and a consumption rate of 0.29 mg/d/larvae [17]. In the case of PI, positive results were obtained, despite being a treatment never tested before. The superworms managed to survive with a value of 60.9% and a consumption rate of 0.2 mg/d/larvae. It should be noted that although the larvae survived after consuming the synthetic diets, they noticeably lost weight, showing thinness with dark coloration and limiting development to their pupal stage. The latter is only an observation, since it was not the object of this study.

It is known that *Zophobas morio* larvae have the ability to survive starvation periods of up to 3–4 weeks [14], which could suggest that the BR diet may be toxic to superworms, since the results obtained showed a 50% loss of larvae after 15 days of feeding with butyl gum. However, in this work, we did not have a negative inanition control because the objective of the study was focused on the ability of superworms to feed on different synthetic polymers in a short period of time. These results show the adaptive plasticity that these insects have achieved over time, which has allowed them to persist by consuming plastic materials that may be present in nature, confirming part of the hypothesis that these animals can survive after feeding on plastics.

There is a wide variety of insects that have the ability to feed on synthetic polymers, such as expanded polystyrene; however, the larvae of *Zophobas morio* have proven to be the most voracious when it comes to consuming PS, eating two times more than other insects of the Coleoptera order, such as *Tenebrio molitor*, and three times more than the larvae of *Galleria mellonella* of the Lepidoptera order under similar conditions [18]. The mealworm, or *Tenebrio molitor*, has the capacity to consume LDPE in low quantities, with a consumption rate of 0.05 mg/d/larvae [19], unlike the results of this work where the superworms consumed an average of 0.25 mg/d/larvae.

In the identification of microbes, it was possible to isolate a greater number of bacteria in the larvae subjected to the PI diet. This was an unusual result, since these were the ones with the lowest consumption of the plastic substrate (6 mg/larvae) but a high survival percentage (60.9%), which suggests that the larvae can feed on this carbon source. However, it may be that its structure is difficult for the superworm to chew; therefore, using the rubber with another texture may yield better results.

Mitra and Das suggested that the intestinal bacteria that proliferate are dependent on the substrate consumed by the insect, which shows that the results obtained in this work are consistent with the information described, since the microorganisms identified in each of the diets were different, with the exception of *Bacillus cereus*, which was found in LDPE and PI diet-based treatments [20].

There are difficulties in investigating the degradation of PI by microorganisms, since it has been documented that it is a very slow process that requires many bacteria to degrade small amounts of polyisoprene, so it has been little considered in research [21]. However, superworms were able to survive after feeding on this material. Larvae that consumed a diet based on polyisoprene were identified as having intestinal bacteria such as *Citrobacter* sp., *Bacillus cereus*, *Bacillus* sp., and *Kliebsella aerogenes*. Although there are no reports on the behavior and gut microbiota of *Zophobas morio* when consuming this polymer, the identified bacteria, such as *Bacillus cereus* and *Bacillus* sp., may be closely related to polymer degradation in the gut, since in other studies it was recognized that bacteria of the bacillus group are able to adhere and form a biofilm on the rubber [22]. Brandon et al. identified *Citrobacter* sp. as a polyethylene-degrading bacterium; therefore, it may be related to polymer degradation in the intestine [23]. In this study, it was determined that gut bacteria of superworms play an important role in the decomposition of synthetic polymers, recognizing that different bacterial groups proliferate in the intestines to metabolize specific types of polymers, meaning that the bacteria identified in the intestines of superworms were dependent on the plastic used in each treatment. Gut bacteria such as *Enterobacter* sp. and *Hafnia* sp. were recognized from the Styrofoam-fed superworms. Jiang et al. [18] reported that the *Hafniaceae* family is involved in the degradation of PS in the intestines of *Zophobas morio* upon consumption of this plastic material. *Hafnia alvei* also have the ability to proliferate and form biofilm on PS plates [24]. Luo et al. [25] recognized that the amount of Enterobacteriaceae increased significantly in the gut of larvae when their diet was based on PS, so it is considered a crucial bacterium in the degradation of this synthetic polymer.

In the treatment with low-density polyethylene (LDPE), three species were identified: *Enterobacteriaceae, Kluyvera* sp., and *Bacillus* sp. These bacteria have been studied for their degradative potential [26], since *Enterobacteriaceae* and *Bacillus* have managed to form colonies on polyethylene sheets, using them as the only carbon source. In addition, other studies showed that *Bacillus cereus* managed to reduce the size of a 30-micron polyethylene sheet by up to 35.7% [27]. Pivato et al. demonstrated that *Kluyvera* sp. remains in the gut of superworms when fed inorganic polymers, such as LDPE and PS, as well as being identified in more insects with the ability to consume these materials [28]. The set of bacteria identified corresponds to the intestinal bacteria culturable in traditional culture medium (LBA), which means that with different microbial culture methods, even more intestinal microbial strains potentially involved in the biodegradation of synthetic polymers can be recognized.

The FTIR spectra suggested that PS was mostly degraded in the intestines of the larvae when the antibiotic was not present. For the sample of larvae treated with the antibiotic, much similarity with the control was seen, identifying the C-H stretching bond at 3000 cm^−1^, the C=C stretching bond at 1450 cm^−1^, and the vibrational bands in the 700 to 800 cm^−1^ range, characteristic of PS [29].

The characteristic bonds of LDPE were recognized in the fecal samples of the larvae when they were not treated with antibiotics, recognizing tension peaks of the C-H bond at 2920 cm^−1^ and C-C at 1450 cm^−1^ [26], in addition to the CH_2_ bending movement at 750 cm^−1^. However, a degradation feature was recognized, since the spectrum associated with the C-O bond at 1100 cm^−1^ was incorporated, indicating oxygen incorporation on the structure [26]. Peng et al. demonstrated that the plastics ingested by the larvae were metabolized, being oxidized during the biodegradation process [27]. Brandon et al. also mentioned that the feces of *Tenebrio molitor* larvae were shown to have oxygen incorporation, forming O-C bonds and alcoholic groups after ingestion of these plastics [23]. The working samples were compared with the organic matter control, and this was shown to have a strong C-H bond signal corresponding to the aliphatic structures of the organic matter [30].

The scanning electron microscopy images showed noticeable differences between the feces of insects fed with PS and those fed with organic matter. Different microbial groups were observed on the surfaces of the two samples; in addition, degradation products with hexagonal shapes were recognized in the case of PS, different from the images reported in the study of the feces of *Galleria melleonella* larvae fed with PS [31], where only a breakage of the polymer structure was observed in the feces. In the control, a homogeneous structure was observed on the entire surface, with spheres completely surrounding the frass fragments.

The degradation of synthetic polymers in the gut of superworms is still unknown; therefore, the results obtained in this research are promising. The hypothesis was confirmed, and related work is expected to continue in order to further investigate the metabolism of insects that have adapted to survive by consuming plastics. Future related studies may focus on the functional and structural knowledge of microbial genes through transcriptome, metagenomics, or proteomics studies, which will facilitate the identification of enzymes associated with the plastic degradation process [32].

## 5. Conclusions

*Zophobas morio* larvae were able to feed on 3 of the 4 types of petroleum-derived synthetic polymers and survive during the 30-day experiment. Nine culturable strains of intestinal bacteria were recognized from the superworms that consumed plastics. Chemical analysis showed signs of degradation of the plastic material in frass, since when the feces of the different treatments with and without antibiotic pretreatment were compared in the chromatograms, the depletion of functional groups characteristic of the synthetic polymers was observed. The SEM images showed concrete differences in the structures of the feces when comparing the feces of larvae fed with PS, LDPE, and the untreated polymer control. These results demonstrated the potential of superworms to feed on synthetic polymers with a biological complex that allows them to metabolize the waste and obtain energy from it. In summary, the obtained results pave the way for further applied research on the use of integrated biological systems focused on the degradation of petroleum-derived compounds.

## Figures and Tables

**Figure 1 microorganisms-12-02118-f001:**
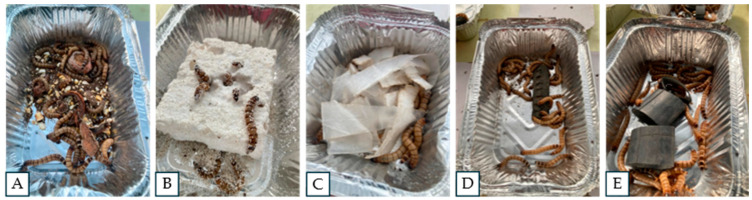
General view of the feeding experiment of *Zophobas morio* larvae with diets of polymers derived from petroleum. (**A**) Control organic mixture, (**B**) expanded polystyrene (PS; Styrofoam), (**C**) low-density polyethylene (LDPE; plastic bags), (**D**) polyisoprene (PI), and (**E**) butyl rubber (BR).

**Figure 2 microorganisms-12-02118-f002:**
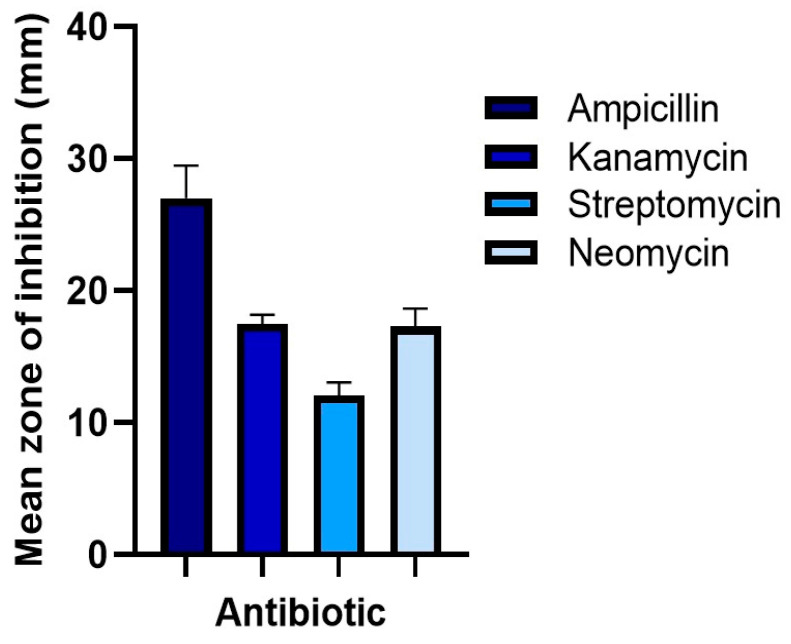
Comparison of the effectiveness of antibiotics in inhibiting the growth of culturable intestinal bacteria.

**Figure 3 microorganisms-12-02118-f003:**
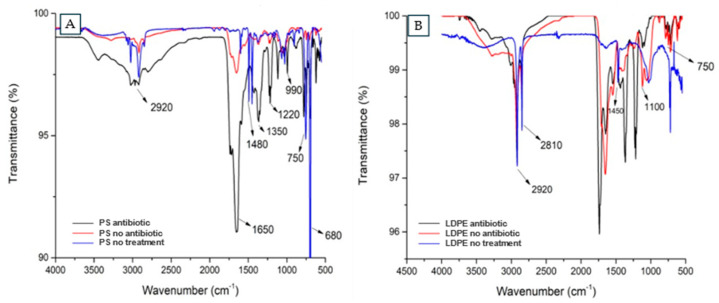
(**A**) Comparison of FTIR spectra in feces of larvae pretreated with antibiotics and fed with PS (PS antibiotic), feces from larvae without pretreated antibiotics and fed with PS (PS no antibiotic), and the negative control of the PS polymer without treatment (PS no treatment). (**B**) Comparison of FTIR spectra of feces from larvae pretreated with antibiotics fed with LDPE (LDPE antibiotic), feces of larvae without pretreated antibiotics fed LDPE (LDPE no antibiotic), and the negative control of LDPE polymer without treatment (LDPE no treatment).

**Figure 4 microorganisms-12-02118-f004:**
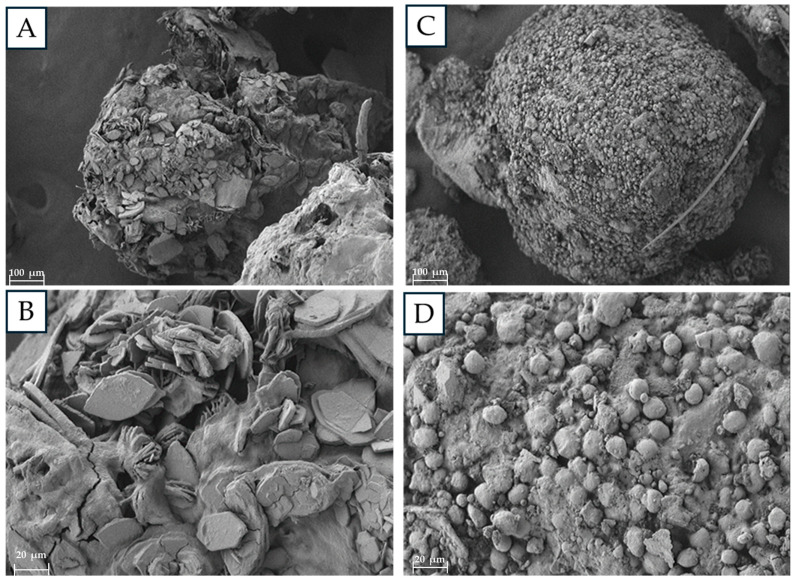
Structures of *Zophobas morio* frass fed with PS and organic substrates. (**A**) PS-fed larvae frass (100 μm), (**B**) PS-fed larvae frass (20 μm), (**C**) organic-fed larvae frass (100 μm), and (**D**) organic-fed larvae frass (20 μm).

**Table 1 microorganisms-12-02118-t001:** Feeding *Zophobas morio* larvae with different diets based on plastics derived from petroleum.

Parameter	Control	PS	LDPE	PI	BR
Alive larvae (number)	28 ± 2 ^a^	16 ± 4 ^b^	20 ± 3 ^ab^	18 ± 3 ^b^	3 ± 1 ^c^
Survival rate (%)	93 ± 8.6 ^a^	53.5 ± 6.2 ^b^	67.4 ± 7.5 ^b^	60.9 ± 6.3 ^b^	10.3 ± 4.1 ^c^
Substrate consumption (%)	100 ± 0.0 ^a^	52.4 ± 7.8 ^b^	4.2 ± 1.9 ^c^	2 ± 0.9 ^c^	0.5 ± 0.1 ^d^
Consumption rate ^1^	11.8 ± 4.3 ^a^	3.2 ± 1.1 ^b^	0.25 ± 0.13 ^c^	0.22 ± 0.15 ^c^	0.06 ± 0.02 ^d^
Weight gain ^2^	+78 ± 50 ^a^	−14.6 ± 4.6 ^b^	−13.9 ± 3.7 ^b^	−17.0 ± 7.9 ^c^	−29.7 ± 5.0 ^d^

^1^ mg/d/larvae; ^2^ mg/survival larvae. Different letters show significant differences according to Tukey’s test (5%).

**Table 2 microorganisms-12-02118-t002:** Cultivable bacterial strains identified in the intestines of *Z. morio* fed on diets based on synthetic plastics and organic matter (vegetable mix).

Treatment	Strain	Identity (%)
PS	*Enterobacter* sp.	99.36%
	*Hafnia* sp.	98.06%
LDPE	*Kluyvera* sp.	96.54%
	*Bacillus* sp.	99.44%
	*Enterobacteriaceae*	99.83%
PI	*Citrobacter* sp.	99.78%
	*Klebsiella* sp	99.08%
	*Bacillus* sp.	98.25%
	*Bacillus* sp.	99.79%
Control	*Klebsiella* sp.	99.52%
	*Pseudocitrobacter* sp.	98.68%
	*Bacillus* sp.	98.94%
	*Bacillus* sp.	99.35%
	*Lysinibacillus fusiformis*	100%

## Data Availability

The original contributions presented in the study are included in the article, further inquiries can be directed to the corresponding author.

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
