# Peer review of "Microbiological and Physicochemical Approach in the Feeding of Superworm (Zophobas morio) with Petroleum-Derived Polymer Diets"

_microorganisms, 2024, doi:10.3390/microorganisms12112118_

Round 1

Reviewer 1 Report

Comments and Suggestions for Authors

The increasing amount of waste containing synthetic polymers poses a major threat to the natural environment. The subject of this paper is undoubtedly very interesting, as the authors presented the results of research on the ability of Zophobas morio larvae to use several types of petroleum-derived polymers as the sole source of food, some of which, such as polyisoprene and butyl rubber, have not been previously studied in this respect. Nevertheless, I have some comments regarding the scope of the research conducted and the value of the presented results.

In the first stage of the study, Zophobas morio larvae were fed with four types of synthetic polymers. The control group consisted of larvae fed with organic matter. In my opinion, such an experiment should include another control group of insects that were not given any food. Zophobas morio larvae can survive for a very long time under starvation conditions, and therefore the survival rate of this group of larvae would allow for a better interpretation of the survival rates of the remaining groups. For example, larvae fed with BR showed a very low survival rate. Comparing the survival rate of this group with the group of starved larvae would allow for determining whether the low survival rate is only the result of a lack of appropriate food or whether BR has a toxic effect on these insects. The presented results of this experiment also lack one very important parameter, i.e. weight change of the larvae after 30 days. This is a parameter routinely given in such experiments, allowing for a better assessment of the ability of the tested species to use a given substrate as a food source.

In the next stage, intestinal bacteria were isolated and identified. There are serious doubts about the methodology used. A small number of bacterial strains were isolated, which were considered to be "associated with the decomposition of synthetic polymers". On what basis did the Authors reach this conclusion? Bacteria were isolated on LB Agar, which is a non-selective nutrient culture medium. Nine strains were indeed isolated from the guts of larvae fed with synthetic polymers, but even in such a case, many studies have shown that the gut microbiota gut of such insects is very diverse. There is no certainty that these few isolated strains have the ability to decompose the synthetic polymers used. It is also intriguing that only 15 bacterial strains were identified from the guts of 20 larvae (five larvae from each treatment). If three isolates were collected from each gut, 60 strains were obtained. Does this mean that the same taxa were identified multiple times? From agar plates on which bacteria from the guts of 5 larvae fed with PS were inoculated, 15 bacterial colonies were collected, as I understand it. Were all of them identified as Enterobacter sp. or Kluyvera ascorbata? If so, this would indeed indicate that these bacteria were quantitatively dominant in the gut of all five larvae. This may suggest that these species are responsible for the decomposition of polymers given as food. The same is true for the other isolated strains. However, there is no certainty in this respect, because other, unidentified bacteria present in the gut may be responsible for the decomposition of these polymers, and the remaining bacterial species may use the resulting organic substances. A weakness of the conducted studies is that only 3 isolates were identified from each larva. With a larger number, the data obtained would be much more statistically significant. However, even with a small amount of collected data, it is worth providing in the Table, for each identified taxon, how many times it was identified. This could be considered a partial microbiological characterization of the gut of the studied insects, although much more valuable results would be provided by metagenomic analysis.

The presented studies also examined the effect of antibiotics. Ampicillin was selected for further experiments out of several. However, no results were presented that justified such a choice. In my opinion, it would be worth presenting the obtained results of isolates' sensitivity to antibiotics, especially since the work does not contain many research results. The selected antibiotic was then used in insect cultures aimed at obtaining feces for FTIR and SEM analyses. Since significant differences in FTIR spectra were obtained, it can be assumed that the activity of bacteria decomposing PS and PBD was inhibited by ampicillin, which resulted in the formation of fewer organic decomposition products available to insects. This would be confirmed by the results of survival rate and substrate consumption, analogous to those in Table 1. Ideally, together with changes in larval weight.  If such data was collected, it should definitely be included.

Comments on the Quality of English Language

Minor editing of English language required.

Author Response

Dear referees (please see all the changes in supplementary material in yelllow) .

We appreciate your critical comments, which we analyzed as a whole, and initially resulted in a modification of the title, where the word "characterization" is replaced by "approximation." With this, our results are more focused and we understand that they can respond to the criticals:
1- Lack of control of larvae without feeding: R/ it is certainly known that they can resist long periods of time, even with predatory habits, however, this work aims to lay the foundations for future commercial applications where such long periods of time are not required, this was explained in the revised document.

2-...lack one very important parameter, i.e. weight change of the larvae after 30 days. R/ this variable was estimated and added to the table and therefore considered in both results and discussion.

3- There are serious doubts about the methodology (isolation) used. The wording was improved to indicate that our objective was to work only with cultivable bacteria, which also supports the change to "approach" since microbiological characterization can give the idea that all microorganisms were studied, which was not the case. Only cultivable strains were studied, discarding those that were repeated, and the use of the antibiotic is understood to be only to control a part of the intestinal microbial community, but that it would be very useful for future work on standardization of biological systems for plastic degradation.

Reviewer 2 Report

Comments and Suggestions for Authors

This manuscript explores the phenomenon and underlying mechanisms of Zophobas morio larvae feeding on and digesting plastics. The authors analyze this from the perspective of gut bacteria and preliminarily confirm the larvae’s ability to digest plastics through SEM. The research findings are interesting, but there are some minor issues to address, including: 1) Some unclear aspects of the methodology need revision; 2) It would be better to add an experiment evaluating whether bacteria were successfully eliminated after antibiotic treatment (the authors may have done this experiment, but the results are not presented); 3) The research background needs more introduction to gut bacteria;

Other specific suggestions are as follows:

Lines 47-48: This data was reported almost 10 years ago. Are there any updated figures?

Lines 54-72: Merge the three paragraphs into one.

Line 72: The introduction to gut bacteria is too brief with just one sentence. It’s suggested to add more description and cite relevant literatures, such as: doi.org/10.1038/s41522-023-00435-y

Lines 73-88: Merge the two paragraphs into one.

Line 76: Use abbreviations after first appearance.

Line 149: The authors should evaluate whether the gut microbiota has been successfully eliminated.

Line 162: More details are needed for the SEM procedures.

Line 184, Figure 1: A scale bar is missing.

Table 2: No need to italicize “sp.” An identity rate below 99% is too low; try to find a bacterial species with a higher identity rate, or name it as “sp.”

Lines 137-141: What’s the reason for conducting this part of the experiment? Where are the results?

Line 249: Where is the control group image? Also, how do the authors determine that this is a plastic degradation product?

Lines 281-285: This part and the content of the next paragraph are both too brief to form independent paragraphs.

Line 303: A typo, should be “in this study”

Line 353: Yes, omics can indeed help further deepen the study of mechanisms. However, this sentence needs references, such as doi: 10.1111/1749-4877.12633

Author Response

Dear referees (please see changes in supplementary material in yellow)

We thank you for your critical comments and remarks, which were all considered and modified in the revised document.

Round 2

Reviewer 1 Report

Comments and Suggestions for Authors

I accept the corrections introduced to the work by the Authors and consider them sufficient. However, I suggest changing the Y-axis label in Figure 2 to "Mean zone of inhibition (mm)".

Author Response

Critical. I suggest changing the Y-axis label in Figure 2 to "Mean zone of inhibition (mm)"

A/. The change was made, thanks for the suggestion

Reviewer 2 Report

Comments and Suggestions for Authors

I am quite surprised that the author did not write a point-by-point response but instead highlighted large sections of the text in yellow directly in the manuscript. It is really difficult to see where the modifications were made. I read through the entire document, and most of it has indeed been revised. I have a small question: I recommended an SCI reference for the author at Line 390, but the author chose a non-English reference instead. Why is that? The previous comment was: Line 353: Yes, omics can indeed help further deepen the study of mechanisms. However, this sentence needs references, such as doi: 10.1111/1749-4877.12633."

Author Response

Crirical. I recommended an SCI reference for the author at Line 390, but the author chose a non-English reference instead. Why is that? The previous comment was: Line 353: Yes, omics can indeed help further deepen the study of mechanisms. However, this sentence needs references, such as doi: 10.1111/1749-4877.12633."

A/ The change was made including the reference recommended by the reviewer, thank you very much